# Application of CRISPR/Cas Systems in the Nucleic Acid Detection of Infectious Diseases

**DOI:** 10.3390/diagnostics12102455

**Published:** 2022-10-11

**Authors:** Junwei Li, Yuexia Wang, Bin Wang, Juan Lou, Peng Ni, Yuefei Jin, Shuaiyin Chen, Guangcai Duan, Rongguang Zhang

**Affiliations:** 1International School of Public Health and One Health, First Affiliated Hospital of Hainan Medical University, Haikou 570102, China; 2Department of Epidemiology, College of Public Health, Zhengzhou University, Zhengzhou 450001, China

**Keywords:** CRISPR/Cas system, biosensing technologies, pathogen nucleic acids, SARS-CoV-2, diagnoses

## Abstract

The CRISPR/Cas system is a protective adaptive immune system against attacks from foreign mobile genetic elements. Since the discovery of the excellent target-specific sequence recognition ability of the CRISPR/Cas system, the CRISPR/Cas system has shown excellent performance in the development of pathogen nucleic-acid-detection technology. In combination with various biosensing technologies, researchers have made many rapid, convenient, and feasible innovations in pathogen nucleic-acid-detection technology. With an in-depth understanding and development of the CRISPR/Cas system, it is no longer limited to CRISPR/Cas9, CRISPR/Cas12, and other systems that had been widely used in the past; other CRISPR/Cas families are designed for nucleic acid detection. We summarized the application of CRISPR/Cas-related technology in infectious-disease detection and its development in SARS-CoV-2 detection.

## 1. Introduction

Infectious diseases, causing a massive burden of disability and death, have always been terrifying threats to humans [1]. Zika virus [2], Ebola virus [3], severe acute respiratory syndrome coronavirus (SARS-CoV), and Middle East respiratory syndrome coronavirus (MERS-CoV) [4], as well as the ongoing outbreak of the SARS-CoV-2 epidemic, has caused great harm to human beings [5]. Fast, accurate, and cost-effective detection and identification of pathogens are key to the control and management of infectious-disease epidemics. According to the World Health Organization, the ideal pathogen-detection method should be fast, specific, sensitive, and not require too-large equipment. The traditional method for pathogen detection was plate culture, which often takes a long time and effort. Following the emergence of real-time polymerase chain reaction (PCR), people can accurately identify pathogenic pathogens through pathogen-specific biomarkers and greatly shorten the detection time. However, real-time PCR also has some limitations, such as requiring complex infrastructure and skilled operators, which limits its deployment in countries with relatively backward medical standards.

The clustered regularly interspaced short palindromic repeats (CRISPR)/CRISPR- associated protein (Cas) system [6] is a sensitive and specific biological system that can quickly identify pathogen-specific nucleic acids [7]. CRISPR systems are present in approximately 90% of archaea and 48% of bacterial genomes and are essential for their genetic components [8,9]. They play an important role in immune function and protect the host against foreign genes [10]. In CRISPR systems, foreign mobile genetic elements (MGE) are firstly cleaved by synthesized corresponding proteins to generate a spacer, then integrated into the CRISPR array; this is the acquisition stage; the second stage is CRISPR RNA (crRNA) biosynthesis. The CRISPR array transcribes and processes the biosynthesis of mature crRNA [11]; the last is the interference stage; the invading MGEs are degraded after being recognized by crRNA [12]. CRISPR systems also show a corresponding diversification in both their Cas protein sequence and species [13]. According to the design principles of the effector module, all CRISPR systems fall into two distinct categories: class 1 systems utilize a large multi-Cas protein complex for crRNA binding and target sequence degradation, while class 2 systems employ a single DNA endonuclease to recognize double-stranded DNA (dsDNA) substrates and cleave each strand with a distinct nuclease domain (HNH or RuvC) [13,14]. CRISPR systems brightly released their potential in various field diagnosis and genotyping applications and brought a leap forward developing opportunities in the field of pathogen detection [15,16].

Some CRISPR systems have been developed to detect nucleic acids or pathogen biomarkers. CRISPR-based biosensing platforms are expected to revolutionize the diagnosis of pathogens [17,18]. The field of CRISPR system exploration is growing rapidly. More and more CRISPR systems are being discovered and applied in nucleic acid detection [19,20]. In this paper, we focused on the more comprehensive application of CRISPR systems in biosensing platforms and showed examples of the detection of the nucleic acids of various pathogens.

## 2. Classification of CRISPR Biosensing Systems

CRISPR systems are a universal immune mechanism. Due to this universal adaptability, CRISPR systems are indeed as diverse as the innate immune system. The Cas protein sequence and genome organization of the CRISPR-Cas locus show great diversity. All CRISPR systems are divided into two different categories [21]. Class 1 systems have multi-subunit effector complexes containing multiple Cas proteins, while in class 2 systems, the effector is a single large multi-domain protein. The two CRISPR-Cas types are further divided into six types. Class I includes type I, III, and IV, and class II comprises type II, V, and VI. Each type is characterized by a different structure of effector modules.

The Cas effector can be a single Cas protein or a large multi-Cas protein complex; though each established CRISPR-based nucleic acid biosensing system has a different combination of components, the fundamental difference is that they use different Cas effectors. For example, Cas9 contains two nuclease domains, RuvC and HNH, which cleave the target and non-target strands of dsDNA, respectively. CRISPR/Cas12, CRISPR/Cas13, and CRISPR/Cas14 have collateral strand cleavage activity, that is, non-specifically cleaving the surrounding nucleic acid molecules after recognizing the target sequence, and CRISPR/Cas10 will trigger its nuclease activity after binding to a perfectly matched target sequence. At present, the most widely used CRISPR systems, such as CRISPR/Cas9, CRISPR/Cas12, CRISPR/Cas13, and CRISPR/Cas14, belong to the type II CRISPR system, and their effector template is a single protein. These systems are easy to be edited for nucleic acid detection [22]. Recently, many researchers have conducted in-depth research on CRISPR/Cas10, which belongs to the type III CRISPR system, and developed some high-sensitivity detection platforms. CRISPR/Cas10 uses RNA as a target sequence for identification and performs well in the diagnosis of SARS-CoV-2 [23]. The reported CRISPR-systems-based biosensing system can be divided into five categories (CRISPR/Cas9, CRISPR/Cas12, CRISPR/Cas13, CRISPR/Cas14, and CRISPR/Cas10). Table 1 lists the main features of the various CRISPR-related detection systems.

## 3. CRISPR/Cas9 Systems

The CRISPR/Cas9 system (Figure 1A), a typical class 2 type II CRISPR system, is widely applied at present [51]. In 2013, the type II CRISPR system was first isolated from Streptococcus pyogenes (SpCas9) and successfully performed RNA-guided DNA cleavage in mammalian cells, paving the way for the CRISPR/Cas9 system as a widely available genome editing tool [52,53]. Under natural processes, the trans-activating crRNA (tracrRNA) base pairs with the repeat sequence in the crRNA to form a unique dual RNA hybrid structure guide that directs Cas9 to cleave the target DNA, so a chimeric sgRNA was designed that combines crRNA and tracrRNA into a single RNA transcript, simplifying the system while preserving Cas9-mediated full-function sequence-specific DNA cleavage [14,54]. Cas9 contains two nuclease domains, RuvC [55] and HNH [56], which cut the target DNA strands and non-target DNA strands respectively [57]. A short trinucleotide protospacer adjacent motif (PAM) is also essential for initial target sequence recognition [58]; the target sequence could not be recognized without a corresponding PAM site. After successful identification, a double-strand break (DSB) occurs upstream of the 3′-NGG PAM site. Currently, the CRISPR/Cas9 system has been widely used in genome editing, single-nucleotide-mutation detection, and other fields [59,60,61,62].

### 3.1. CRISPR/Cas9-Based Biosensing Systems

Nucleic acid sequence-based amplification (NASBA)-CRISPR Cleavage (NASBACC) [24] (Figure 2A), based on the CRISPR/Cas9 system, was designed to detect the Zika virus. First, the reverse primer of NASBA was used to amplify the sample RNA sequence [63], and a specific sensor trigger sequence was added to the amplified product. CRISPR/Cas9 was introduced to achieve a specific cleavage reaction. When the target sequence is present, the dsDNA synthesized by the NASBA reaction will be cleaved by CRISPR/Cas9, and the truncated RNA product will not activate the sensor switch. However, if no target sequence existed, the added sensor sequence can trigger a subsequent reaction to cause color changes and finally achieve detection. In the Cas-exponential amplification reaction (EXPAR) [25] (Figure 2B), the Cas9/sgRNA complex mediates the site-specific cleavage of ssDNA substrates, produce cleaved fragments (X). X hybridizes to the EXPAR template and is extended along the template by DNA polymerase from its 3′ end. The subsequently formed duplex is cleaved by Nease; a copy of X is released from the template, and then the dissociated X continues the same process as above, serving to amplify the signal. At the same time, the mixture was incubated in a real-time PCR assay system, and the fluorescence intensity was monitored at 1 min intervals, with a detection limit of 0.82 Attomolar (aM). Cas-EXPAR could be used for site-specific DNA methylation detection by combining it with bisulfite conversion. There are also CRISPR/Cas9-based detection systems designed with the UiO66 platform [28] (Figure 2C). UiO66 is a nanoporous material [64]. The fluorescence of ssDNA is quenched in the metal–organic framework platform based on UiO66 [28]. Two Cas9/sgRNA ribonucleoprotein complexes were designed to recognize and cleft the target DNA to produce short ssDNA. Then the circular probe hybridized with the short ssDNA to form long repetitive ssDNA by rolling loop amplification [65]. In the presence of long-ssDNA, the fluorescent probe leaves UiO66 and hybridizes with long-ssDNA, leading to fluorescence recovery. Therefore, the quantitative detection of target DNA can be evaluated by the recovered fluorescence intensity.

Some platforms achieve nucleic acid detection through the sequence-specific binding ability of inactivated Cas9 effector [66]. Paired dCas9 (PC) reporter system [29] (Figure 3A), designed two paired dCas9/sgRNA ribonucleoprotein complexes that are respectively connected to the N-terminal and C-terminal of the firefly luciferase (NFluc and CFluc) [67]. Upstream and downstream sequence segments are adjacent; when these two fragments are present, two dCas9/sgRNA ribonucleoprotein complexes combined and activated luciferase activity, finally generating luminescence for detection. The VirD2(virulence D2)-dCas9 guided and LFA-coupled nucleic acid test (Vigilant) [27] (Figure 3B), combining CRISPR/Cas9 with lateral flow analysis (LFA) technology, designed a direct-detection platform. The deactivated Cas9 of Streptococcus pyogenes (spdCas9) was fused with VirD2 relaxase [68] and designed as a reporter ssDNA with the 5′-end of the 25-BP VirD2 recognition sequence and the 3′-end of the biotinylated sequence. When the target nucleic acid sequence was presented, the biotin-reporting ssDNA-VirD2-Cas9-sgRNA-targeting ssDNA complexes formed and finally reported results by LFA [69].

Recently, CRISPR/Cas9 has also been integrated into sequencing systems, such as finding low abundance sequences by hybridization–next-generation sequencing (FLASH-NGS) [26] (Figure 3C). In this system, the combination of recombinant Cas9 and multiple-guide RNAs can accurately remove unwanted background sequences [70] and also can target almost any interest sequence without optimization. The mechanism was to seal the sample genomic DNA or cDNA with phosphatase, then digest it through multiple combinations; each sgRNA-guided Cas9/sgRNA ribonucleoprotein complexes can cut the interest sequence into Illumina sequencing-size fragments, finally generating a cleavage product that can be connected to a universal sequencing connector. Through subsequent amplification, the target sequence was enriched and combined with sequencing flow cells to achieve multiple detections. This method shows good sensitivity in highly multiple detections of antibiotic genes and has certain guiding significance in the development of cancer-mutation detection, rare mosaic allele detection, and the targeted transcriptomics of clinical samples.

### 3.2. Evaluation of CRISPR/Cas9-Based Biosensing Systems

The detection platform based on CRISPR/Cas9 has brought new opportunities to the detection of infectious diseases, but there are still large deficiencies in the whole system. At first, it can allow up to six consecutive mismatches in the 5′-end region of the prototype gasket, which greatly increases the occurrence of off-target effects. Second, it requires 3′GC-rich PAM for anchoring, which has some limitations in selecting target gene segments for detection.

In contrast to RT-PCR, NASBACC uses a paper-based sensor platform to advance field-ready diagnostics, but even when strains of two different lineages are mismatched up to 4-nt (11%), it can also fully activate the sensor, limiting its value in practical applications. Cas-EXPAR does not require exogenous primers for detection, avoiding target-independent amplification, and can be used as a general method for the detection of DNA, RNA, methylated DNA, and other nucleic acids, with high specificity and rapid amplification kinetics, which has great application potential in biological analysis and disease diagnosis. The UiO66-platform-based Cas9 provides significant sensitivity under mild reaction conditions and has good selectivity for different pathogens. It also shows good performance in actual sample analysis, but it is three orders of magnitude lower than the detection limit of RT-PCR. Different MOF materials are the main influencing factors of fluorescence quenching to recover efficiency. Finding better materials may be beneficial to further reduce the background signal and improve the sensitivity of the detection system. The use of luciferase as a reporter gene in the PC reporter system could be used for the rapid prototyping of arrays and microsystems in the future, but for field applications, electrochemical signals, and colorimetric readings may be more suitable. Key features of Vigilant include short run times, compatibility with rapid extraction protocols, and isothermal amplification, which make it a practical method for detecting viruses and pathogens that can be used for the large-scale screening of COVID-19 cases, but the identification of nucleic acid sequences is dependent on PAM sequences. If any DNA sequence can be identified in a PAM-independent manner, the utility of this platform will be extended to any nucleic acid sequence, and the detection process of pre-amplification and detection step separation and how to better avoid cross-reaction also needs to be further considered. FLASH-NGS enable high levels of multiplexing (thousands of targets) and the highly multiplex detection of antimicrobial resistance genes directly in a patient′s sample, but if the patient is infected with multiple microbial properties, it is not possible to determine whether the acquired resistance genes originated in a particular species.

## 4. CRISPR/Cas12 Systems

CRISPR-Cas12a (Cpf1) (Figure 1B) contains a predicted RuvC-like endonuclease domain, which can cleave dsDNA under the guidance of gRNA; this domain is also closely related to the corresponding nuclease domain of Cas9 [71]. Unlike Cas9, Cas12a recognizes a distal 5′-T-rich PAM and generates PAM distal dsDNA breaks with staggered 5′ and 3′ ends. Cas12a also can recognize complementary ssDNA sequences in a PAM-independent manner and cleave it. Different from Cas9, Cas12 has collateral strand cleavage activity [30,72], as the target DNA sequence is present. Cas12 will release powerful, indiscriminate single-stranded DNA (ssDNA) cleavage activity. Cas12 has been widely developed due to its trans-cleavage activity.

### 4.1. CRISPR/Cas12-Based Biosensing Systems

In simple terms, the Cas12a/gRNA effector is used to identify the target DNA sequence, forming a Cas12/gRNA/target DNA ternary complex and then non-specific trans-cleavage fluorescence quenching reporter probe ssDNA [40,73]; finally, detection results are obtained through changes in the fluorescence signal [31,74]. Like (DETECTR) [30] and HOLMES [31], *Laevis* family bacteria-derived LbCas12a is used for detection by the above principle, and an isothermal amplification step of the sample sequence is also introduced to improve the sensitivity. There is also an ITP–CRISPR assay [32] that uses an electric field gradient to purify nucleic acids and accelerate DNA and RNA hybridization to speed up CRISPR analysis [75,76]. Cas12/gRNA, target DNA, and reporter ssDNA were confocalized into a 100 μL system. Changes in fluorescence intensity were used to report the results. This method was successfully applied to the rapid detection of SARS-CoV-2 RNA, and the total detection time from the original sample to the result was reduced by about 30–40 min, demonstrating its excellent diagnostic ability.

Cas12a-FDet [33] combines the reaction system in a sealed reaction tube, avoiding the risk of aerosol contamination during amplification and transfer. Like Cas12a-FDet, a one-pot toolbox with precision and ultra sensitivity (OCTOPUS) concentrated the recombinase-polymerase amplification (RPA) reagents [34], crRNA, and ssDNA-FQ reporter genes in a single tube, but Cas12a protein was stored on the tube cap. RPA was pre-amplified 15 min after sample collection, and then the reaction tube was rotated to make the Cas12a protein enter the system. After the target sequence was identified, CRISPR/Cas12a was activated, and the fluorescence reporter gene was cleaved to generate fluorescence for detection.

Not only the fluorescence quenching reporter ssDNA probe is used to report the detection result, but some systems use other detection methods, such as Cas12a-UPTLFA, combined with LFA technology and up-converting phosphor nanoparticle (UCP) label-coupled detection methods [35]. The method used a FAM-biotin probe (UPT will be conjugated to biotin), labeled ssDNA as the reporter, and the results were obtained by scanning using a UPT biosensor. The cutoff value was set according to the T/C values of the negative controls to designate the positive or negative results. When the target sequence is present, the ssDNA is cleaved, and the UPT conjugate flows along the flow band, causing a change in the T/C value, effectively improving the detection sensitivity. There also have some CRISPR/Cas system-based detection systems based on electrochemical biosensor platforms [77], which have the advantages of fast signal reading, simple platform, and low transducer cost. Recombinase-aided amplification (RAA)-based E-CRISPR [36] (Figure 4A) uses methylene blue (MB) to modify the ssDNA reporter gene via the Au–S covalent bond and assemble it on the working electrode. First, the sample is amplified by recombinase-aided amplification (RAA). Then the non-specific ssDNA trans-cleavage activity of the CRISPR/Cas12a was activated in the presence of the target sequence. The reporter gene modified by MB was cleaved, and the results were reported by wave voltammetry (SWV) to detect before and after the microelectrochemical signal, to achieve the purpose of nucleic acid detection. The electrical impedance spectrum (EIS)–CRISPR [37] (Figure 4B) fixed ssDNA on the gold electrode, limiting electronic communication between the electrode and the solution. In the presence of the target DNA, CRISPR/Cas12a binds to the target DNA and triggers its trans-cleavage activity, cracking the ssDNA on the gold electrode, accelerating electron transfer between the electrode and the solution. Finally, combined with EIS technology, the presence of the target sequence is identified by detecting subtle changes in the current on the electrode surface [78].

CRISPR/Cas12b [30] has the same non-specific trans-cleavage capability as CRISPR/Cas12a. However, the CRISPR/Cas12b system showed different target preferences during the trans-cleavage process, and the non-specific ssDNA trans-cleavage rate was higher when dsDNA was used as the target than ssDNA. In addition, CRISPR/Cas12b targets ssDNA substrates by cleaving the ssDNA probe independently of the PAM, whereas targeting dsDNA requires the 5′-TTN-3′ PAM site [79]. The one-hour low-cost multipurpose highly efficient system (HOLMESv2) is a nucleic acid detection platform based on the CRISPR/Cas12b system [38]. Cas12b, as a thermophilic Cas protein, can bind to the loop-mediated isothermal amplification (LAMP) program [80]. The optimal PAM sequence (5′-TTC-3′ and 5′-TAC-3′) and the optimal reaction temperature (target ssDNA 35–65 °C target dsDNA 45~55 °C) were discussed, and a one-step detection system was constructed that could be used to distinguish SNPs and accurately quantify the methylation degree of the target DNA. The Cas12b-based DNA detection (CDetection) platform also utilizes the trans-cleavage activity of Cas12b [39], which can simultaneously distinguish between HPV16 and HPV18, with a detection limit of 10nM without pre-amplification.

### 4.2. Evaluation of CRISPR/Cas12-Based Biosensing Systems

These detection platforms based on the CRISPR/Cas12 system mainly rely on their collateral strand cleavage activity, combined with the application of fluorescence quenching reporter genes, so that the detection results can be better presented. However, the specific recognition of target sequences still requires the participation of 5′ AT-rich PAM, and the selection of target sequences is still limited. In addition, the CRISPR/Cas12 system has a high tolerance to the first eight nucleotide mismatches (<4 nt) near PAM, making the off-target effect more inevitable.

HOLMES tested 10 kinds of Cas12a proteins and finally selected the Trichospiraceae bacteria ND2006 Cas12a (LbCas12a) to be used in related research, which laid a foundation for subsequent research. DETECTR can quickly and specifically detect HPV in human patient samples, thus providing a simple platform for point-of-care diagnosis based on nucleic acids, which can be extended to rapidly detect any DNA sequence of interest with high sensitivity and specificity, but its practical application aspects have not been talked about. In ITP-CRISPR, an electric field gradient is used to control and influence the rapid CRISPR-Cas12 enzyme activity when the target nucleic acid is recognized. It takes about 30 min from the original sample extraction to obtain the result. However, due to the limitation of microflow chip design, the analysis of this detection system is currently limited to processing 10 μL original sample as input. This can affect the sensitivity of the system, and the sample cleavage and LAMP steps in the process need to be done manually, which is a hindrance to achieving high-throughput detection. The Cas12aFDet system has certain anti-interference ability in complex sample detection, and the use of constant room temperature as the reaction temperature and a single-tube detection process renders the system expected to be used for immediate detection and also explores a feasible efficient one-pot detection system. OCTOPUS is also a one-pot detection system, which omits the additional opening process to avoid practical inconvenience and possible cross-sample contamination. It can reach its detection limit of 1 CFU/mL in less than 50 min, but its cost is much higher than RT-PCR (7–8 times), making it difficult to use for widespread disease and epidemic control. A portable UPT biosensor is used in CAS12A-UPTLFA. Non-professionals can complete CAS12A-UPTLFA within 1.3 h as a POCT method, and it does not require expensive quantitative PCR instruments or microplate readers. It is versatile and can be prefabricated in batches. However, the successful detection rate at the lowest detection concentration is only 75%, which limits its practical application. RAA-based E-CRISPR converts target-recognition activity into detectable electrochemical signals, which greatly improves sensitivity. However, due to the deficiency of its detection limit, it is necessary to conduct pre-enrichment treatment before the RAA reaction in the detection of low-pollution samples. Whether it can be well combined with the amplification step still needs further exploration. EIS—CRISPR also converts target-recognition activity into a detectable electrochemical signal and can specifically identify different clinical isolates, but the main limitation of this system is its detection response time, whether LAMP can be combined with it, and shortening the time required for detection still needs to be further explored.

## 5. CRISPR/Cas 13 Systems

Cas13a (C2c2) (Figure 1C) belongs to the type VI CRISPR/Cas system [81], contains two nucleotide sequence binding domains (HEPN), and has single-stranded RNA (ssRNA) cleavage activity [82]. In 2016, it was discovered to have additional cleavage activity triggered by target RNA, making it widely used in nucleic acid detection [83].

### 5.1. CRISPR/Cas13-Based Biosensing Systems

CRISPR/Cas13-based biosensing systems include specific and high sensitivity enzymatic reporter gene unlocking (SHERLOCK) [40] and combinatorial arrayed reactions for multiplexed evaluation of nucleic acids (CARMEN) [42]. CARMEN integrated CRISPR/Cas13, reporter gene mixtures, and pre-amplified samples into the microarrays, using fluorescence microscopy to monitor the reaction of each microwell, enabling a high degree of multiple nucleic acid detection. This method can also be used to detect low-frequency cancer mutations in cell-free (cf) DNA fragments, as well as homozygous and heterozygous genotypes based on single-base differences. More importantly, this method has great potential for the large-scale diagnosis of SARS-CoV-2 [42].

An easy-readout and sensitive-enhanced (ERASE) band based on CRISPR/Cas13 is an improved version of the more sensitive lateral flow strip detection technology [43]. Optimizing the number of reporter RNA molecules, streptavidin, and secondary antibodies in the ERASE band, the cleaved reporter molecule could only be detected in the antibody capture band (C-band), and the biotin capture band (T-band) could not detect a cleaved reporter molecule, while the uncleaved reporter molecule could be detected on both bands. The improved lateral flow strip not only has higher sensitivity and specificity but also has more intuitive results. The disappearance of the T-band was regarded as the positive threshold. ERASE has great potential and needs to be further developed. Some researchers try to use the change in turbidity as the result of qualitative diagnoses, such as the liquid–liquid separation phenomenon (LLPS) (the critical value of polymer length exists in the polymer solution; when the polymer length reaches the critical value, the LLPS phenomenon will appear, the solution becomes turbid; when the polymer length is lower than the critical value, the solution components are evenly mixed, and the solution is clear) [84]. Combined with the collateral cleavage activity of Cas12a or Cas13a, the designed LLPS-Cas12a/Cas13a [44] (Figure 5A),cleaves long-chain nucleotides when the target sequence appears, renders the solution clear, and realizes detection through the change of solution turbidity.

Besides the detection of pre-processed nucleic acid sequences, researchers have designed several detection systems that are not limited to pre-processed samples. For example, the light-up RNA aptamer signaling-CRISPR-Cas13a assay [45] (Figure 5B) enables the direct detect of pathogens based on the CRISPR/Cas13 system. This technology introduced a luminescent RNA aptamer in the Broccoli/DFHBI-1T complex [85]. After the aptamer was digested, the free DFHBI-1T only emits weak fluorescence; while the aptamers were combined with broccoli, the fluorescence of DFHBI-1Tis emit was more than 100 times than the free state [45]. Meanwhile, the bacteria RNA was identified by CRISPR/Cas13 to judge whether the pathogen was alive or not and quantifies the target RNA content to accurately quantify the live bacteria. This technology shows good sensitivity and specificity in the detection and quantification of live Bacillus cereus in food samples. Allosteric probe-initiated catalysis and the CRISPR-Cas13a system (APC-Cas) [41] (Figure 5C) requires only a small number of bacteria for detection. An allosteric probe (AP) containing three functional domains (aptamer domain, primer domain, and T7 promoter domain) was introduced [86,87]. The aptamer domain was used to identify specific strains; the primer domain was used to amplify the amplified sequence, and the T7 promoter domain was used to initiate RNA transcription and amplification. AP is inactivated in the absence of the target pathogen, but once the target pathogen is present, the aptamer domain specifically recognizes and binds to it, allowing the AP to expand from hairpin-like inactivated structures to active structures. With the participation of DNA polymerase and primer, dsDNA was generated using AP as a template strand, and then the target pathogen was replaced, and the first amplification was achieved. The secondary amplification was then performed with the participation of T7 RNA polymerase; the ssRNA was then recognized by the Cas13a/crRNA complex, which activated the trans-cutting ability to cleave a large number of surrounding RNA gene reporter probes to achieve the third amplification. Finally, the results were identified in combination with the generated fluorescence signal. The researchers successfully tested the specificity of Salmonella enteritidis using this technique. These two systems do not require the pre-processing of samples, further reducing the detection platform requirements, and are more conducive to on-site detection. The area of technology that combines the CRISPR/Cas system and detects pathogens directly still has great potential.

### 5.2. Evaluation of CRISPR/Cas13-Based Biosensing Systems

CRISPR/Cas13 also has collateral strand cleavage activity, allowing non-specific cleavage around RNA after recognition. Using this direct RNA recognition activity, researchers developed assays that do not require sample pretreatment, which greatly improves the utility of the CRISPR/Cas system. Furthermore, the high-sensitivity recognition of the target RNA sequence can better reduce the impact of the off-target effect.

In the SHERLOCK system, reagents can be lyophilized for long-term storage and reconstituted on paper for field applications at a cost as low as $0.61 per test. However, combining the Cas13a detection reaction with paper-spotting significantly reduced the absolute signal of the readout, and the detection limit was also reduced by a factor of 10. CARMEN enables large-scale CRISPR-based diagnostics, and CARMEN can overcome the challenge of sequence heterogeneity at target sites through crRNA multiplexing, and the costs can be as low as $0.05/test. In the broader context of pathogen detection, discovery, and evolution, CARMEN and NGS complement each other. The ERASE system provides a powerful visualization tool for CRISPR detection and fulfills the possibility of on-site COVID-19 diagnosis with minimal equipment, but the presence of shallow T-bands that do not completely disappear in the result report may lead to some weakly positive samples being judged as negative; the sensitivity is slightly lower than that of precision fluorescence detection equipment. The LLPS-CRISPR system provides a simple and inexpensive way to implement CRISPR-based molecular diagnostics and circumvent the introduction of chemical labels on DNA or RNA molecules, which is more environmentally friendly and less expensive than other methods, but its detection limit is low, and whether the sensitivity can be improved by several orders of magnitude by using an appropriate amplification reaction requires further experimental verification, and evaluating the test results with the naked eye or simple equipment is prone to false negative results. The light-up aptamer-based-Cas13a system uses a one-pot assay for live pathogen detection, capable of detecting target bacteria as low as 10 CFU and accurately quantifying live bacterial content from 0 to 100%; the integration of CRISPR-Cas 13a and light-up RNA aptamers helps to create a reverse-transcription-free, nucleic-acid-amplification-free, and label-free method, further reducing equipment requirements, testing time, and cost. However, since the shutdown response of light-up RNA aptamers to target bacteria will greatly affect the detection of the system, finding more suitable light-up RNA aptamers needs further exploration. The APC-Cas system can easily detect Salmonella Enteritidis levels as low as 1 CFU, far exceeding the detection limit of 400 CFU of RT-PCR, and can effectively distinguish contaminated milk from pasteurized milk; it does not require bacterial isolation, nucleic acid extraction, or washing steps, and it costs as low as $0.86 per test. However, the system uses three enzymes in three steps, which makes the operation process complicated. Further simplifying the detection process without changing the detection sensitivity will greatly enhance its practical application value.

## 6. CRISPR/Cas 14 Systems

The Cas14 protein (Figure 1D) is a low-molecular-weight (400–700 amino acid) RNA-guided nuclease that recognizes target ssDNA and cleaves it without restriction sequences [88,89]. CRISPR/Cas14 also performs non-specific cleavages around ssDNA nuclease molecule like CRISPR/Cas12 [88].

### 6.1. CRISPR/Cas14-Based Biosensing Systems

Some nucleic-acid-detection platforms are based on CRISPR/Cas14, such as CMP [47] and TSPE-Cas14a [46] (Figure 6). In the CMP system, pre-amplified samples are firstly, and the CRISPR/Cas14 system-mediated activation of the unique collateral strand cleavage activity is used to cleave fluorescence quenching ssDNA reporter probes in the presence of the target sequence; then direct fluorescence reading is used for pathogens diagnosis. The tag-specific primer extension (TSPE)-based CRISPR/Cas14a pathogenic detection system designed a tag-specific primer that contains two domains (a primer sequence domain matching the target, and a tag sequence domain matching sgRNA). This assay does not require the purification of nucleic acid; the primer sequence domain amplifies and is enriched after matching the target sequence, and then the target nucleic acid is separated from the mixture by streptavidin-coated magnetic beads (the sequence-specific amplicons containing the tag sequence and biotin tag sequence are captured by streptavidin-coated magnetic beads). Identifying the tag sequence domain matching sgRNA activated the trans-cleavage activity of CRISPR/Cas14 and cleaved the fluorescence quenching reporter gene, resulting in an enhanced fluorescence signal and achieved detection. This diagnostic platform has a low detection limit (single cell or one aM), high sensitivity, and wide adaptability. CRISPR/Cas14 system-mediated non-specific collateral strand cleavage activity does not require a specific PAM sequence; CMP and TSPE-Cas14a can easily detect various DNA targets of pathogens by redesigning sgRNA.

### 6.2. Evaluation of CRISPR/Cas14-Based Biosensing Systems

Cas14a is established as the smallest class 2 CRISPR effector demonstrated to date for programmable RNA-guided DNA cleavage and can be detected without the restriction of PAM sequences. The recognition of ssDNA substrates by Cas14a requires a seed region that is different from the PAM region for complementation, and more than 2-nt mismatches will strongly inhibit the activity of Cas14a. This provides a huge possibility for its realization of high-fidelity detection, which greatly increases its practical application value.

The CMP system provides a good model for the future accurate and direct detection of pathogens, especially single-stranded DNA viruses. Based on the collateral cleavage activity of CRISPR-Cas14a and TSPE, a general bacterial nucleic acid diagnostic platform has been developed, with a low detection limit (single cell or one aM), high accuracy (99%) and wide adaptability. However, this diagnostic platform is very time-consuming. In the future, combining isothermal amplification, lateral flow, and other technologies with Cas-TSPE to develop an easy-to-use integrated diagnostic method is waiting to be explored.

## 7. CRISPR/Cas 10 Systems

The type III CRISPR/Cas system is a multi-component and multi-pronged immune effector, and its reprogramming and use are more complicated due to its complex enzyme structure [90]. Relying on the unique internal signal-amplification mechanism of the type III CRISPR/Cas system [91], the type III CRISPR complex can be programmed to specifically recognize the virus RNA. After recognizing the target virus RNA, the cyclase domain produces about 1000 cyclic nucleotides (cOA); the cOA activate Csx1, cutting off the fluorophore connected to the quencher [23] (Figure 1E). Although these detection methods save time, the detection limits are relatively insufficient. When combined with pre-amplification, the time advantage will be lost to some extent, but it still proves that the type III system can also be used for rapid and sensitive detection.

### 7.1. CRISPR/Cas10-Based Biosensing Systems

Some nucleic acid detection systems based on the type III CRISPR are designed to detect viral RNA, such as MORIARTY [48] (Figure 7A) and SCOPE [50] (Figure 7B). MORIARTY was designed with a recombinant active Lactococcus lactis Csm (LlCsm) complex to synthesize cOA6 with the participation of divalent ions and ATP, and then cOA6 activated the RNase activity of Csm6. Then RNA-FAM was cleaved to cause changes in fluorescence intensity and finally achieve the detection. Compared with real-time PCR, it does not require expensive equipment and can even detect target RNA at 0.5 fM under non-amplification conditions. Combined with reverse transcription and RT-RPA pre-amplification steps, the detection limit can reach the aM level. SCOPE designed the TtCmr/crRNA complex targeting RNA by purifying the endogenous Cmr complex (TtCmr) from T. Thermophilus HB8. Upon the recognition of the target RNA, the complex generates cOA molecules, which then triggers the cleavage of the reporter RNA by TTHB144, resulting in a detectable fluorescence signal. In addition, the target RNA sequence required for TTHB144 activation is identical to the sequence required for cOA synthesis, allowing for the complete distinguishing of single-base differences. Combined with RT-LAMP and other pre-amplified sample methods, the detection limit can also reach the aM level. There is also a VmeCmr–NucC coupled assay that is based on the III-B CRISPR system from Vibrio metoecus, uses purified VmeCmr to activate NucC by cA3 generated during the activation of target RNA, and then cleaves the fluorescent reporter gene to report the detection results [49].

### 7.2. Evaluation of CRISPR/Cas10-Based Biosensing Systems

The nucleic acid detection system based on the type III CRISPR/Cas system is faster and can directly detect RNA samples. Combined with some pre-amplification steps, the detection limit can reach the aM level. However, due to its complex effector protein, its programming is more complicated, and it is rarely used in detection technology and needs further exploration.

SCOPE is the first class 1-based CRISPR-Cas nucleic-acid-detection tool with high sensitivity and specificity, rapid detection, and flexibility. However, the detection system requires an additional amplification step, and the two-step method limits its application to high-throughput testing. Finding an efficient one-pot system can alleviate this situation. The MORIARTY assay also showed detection sensitivity consistent with RT-PCR and appeared to have a greater dynamic range than RT-PCR in detecting viral RNA, with a temperature range requirement of 37–42 °C for all steps, eliminating the need for expensive equipment and making it potentially compatible with low-cost hand-warmer-mediated heating solutions, but whether sample complexity reduces assay sensitivity and how can it be better combined with isothermal amplification steps to boost the signal-to-noise ratio are still waiting to be explored.

## 8. Application of the CRISPR/Cas System in SARS-CoV-2 Nucleic Acid Detection

In recent years, there has been a global pandemic of COVID-19 [92,93]. Many researchers have also tried to use CRISPR/Cas systems to facilitate the detection of SARS-CoV-2 [94,95]. The CRISPR-based system not only has better detection sensitivity, but also has great advantages in various aspects such as detection threshold, shortening the detection time, improving the detection efficiency, and reducing the consumption of detection reagents [95,96].

### 8.1. CRISPR/Cas9 Systems in SARS-CoV-2 Detection

Vigilant [27], based on CRISPR/Cas9, designed the sgRNA that targets the SARS-CoV-2 N gene. The detected LOD can reach 2.5 copies/μL, and the sensitivity and specificity are 96.4% and 100%, respectively. This method can achieve detection in 35 min and can be widely used in areas with relatively scarce resources, after pre-assembly with isothermal amplification.

### 8.2. CRISPR/Cas12 Systems in SARS-CoV-2 Detection

The ITP-CRISPR assay is a detection system based on CRISPR/Cas12 [32]. In the rapid tests of SARS-CoV-2 RNA, the virus N and E genes are tested as target sequences (when N or E genes were detected, the test result was interpreted as positive), with sensitivity and specificity of 93.8% and 100%, respectively. This method uses less than 0.2 μL of reagents for detection and can achieve automatic nucleic acid extraction, accelerate, and enhance CRISPR enzymatic reaction for detection; the total detection time can be reduced to 30 min. Compared with real-time PCR, which needs 30 min to one hour to extract nucleic acid and which has a total detection time of nearly 3 h, the ITP-CRISPR assay has a huge advantage.

### 8.3. CRISPR/Cas13 Systems in SARS-CoV-2 Detection

The CRISPR/Cas13 system is also be applied in SARS-CoV-2 detection. CARMEN [42] was a high-throughput detection system designed based on a microarray chip. In a test of 400 SARS-CoV-2 samples, its detection sensitivity could reach 99.7%. There is no doubt that its high-throughput detection capabilities can play a huge role in large infectious diseases such as SARS-CoV-2. The CRISPR/Cas13-based ERASE [43] is simple, fast, cheap, and convenient. Strip brightness was used as the identification standard, and the nucleoprotein (N) gene of SARS-CoV-2 was used as the target sequence. The sensitivity and specificity of the method were 90.67% and 99.21%, respectively, in 649 clinical samples. Although the visual measurement is slightly less sensitive, it is still very suitable for nucleic acid detection in a large population to facilitate the tracing of the source of infections [43].

### 8.4. CRISPR/Cas10 Systems in SARS-CoV-2 Detection

The type III CRISPR/Cas system is also designed to detect SARS-CoV-2. For example, MORIARTY [48] selects the region segment within the spike (S) gene of the SARS-CoV-2 virus as the target RNA, while ensuring that the 3′-protospacer flanking sequence (3′-PFS) of the viral target RNA remained with the 5′-tag of LlCsm crRNA. The temperature range of all steps in the test is required to be 37 to 42 °C, which avoids expensive equipment and improves its on-site utilization value. SCOPE [50] designed a crRNA that targeted the E gene of SARS-CoV-2. Even target RNA at 1 nM can be detected through this system, and the fluorescent signal can be detected within a few seconds after incubation. After LAMP was pre-amplificated, sensitivity detection reached an aM level (10–18 M) in about 35 min (30 min pre-amplification + 5 min CRISPR reaction). The target RNA sequence required for activating TTHB144 needs to be highly consistent with that required for cOA production, thus reducing the false-positive rate. In addition, the temperature required for all detection steps is around 55 °C, which is also compatible with commercially available RNA polymerases. In a VmeCmr–NucC coupled assay, the detection limit can reach 8 fM when the SARS-CoV-2 N gene is used as the target sequence, and the determination is carried out at 37 °C, which has guiding significance for the field population screening of COVID-19.

## 9. Perspective

The CRISPR/Cas system has attracted more and more attention since it was found to have specific recognition of target DNA or RNA sequences and/or non-specific collateral strand cleavage activity. CRISPR-based diagnostic systems work primarily by altering the gRNA to identify any desired target sequence. The combination of biosensing technology with CRISPR not only improves the sensitivity and specificity of existing detection technologies but also greatly reduces the time and cost, shortening the pathogen detection process. The diagnostic system is simple in sample processing, easy to operate, not limited by various environments, does not need special instruments or expensive reagents, and is expected to establish a wide range of real-time diagnosis platforms [97].

The detection capabilities of CRISPR also have certain limitations that have hampered its development. Firstly, detection based on sequence can not avoid interference from the off-target effect. For example, when using CRISPR/Cas9/sgRNA to identify the target sequence, mismatches in the proximal region of PAM were highly tolerated [98], and single mismatches in the interval region were fully tolerated by LshC2c2 [83]. Of course, there are also high-fidelity CRISPR/Cas systems developed for nucleic acid detection, such as HypaCas9 developed in 2017, which greatly improves the ability of targeted detection and reduces the off-the-target effect [99]. Future research should also develop in the direction of high-fidelity detection. Secondly, CRISPR-based nucleic acid detections are mostly qualitative results, unable to determine the pathogenic nucleic acid load of patients, and the pathogenic nucleic acid load is often closely related to the patient′s disease development. Further understanding of the load of the target pathogen can help predict the development stage of the disease and provide guidance for subsequent treatment. How to better combine quantitative detection with CRISPR is also the focus of future research. In addition, there are also problems of sample cross and aerosol contamination in the detection process. Measures such as a centralized detection reaction system in a tube or the pre-storage of the Cas protein in the tube cap can certainly reduce cross-contamination to a certain extent, but these measures are difficult to achieve in high-throughput detection systems [42]. Combining high-throughput testing with less cross-contamination is another challenge.

In summary, CRISPR-based biosensing technology is a major innovation in detection technology that has already had an impact on detection and diagnostic capabilities in many areas and will have an even greater impact in the future. In the face of current and future infectious-disease outbreaks, CRISPR-based biosensing could dramatically improve our ability to diagnose and perform mass screening in populations. The prevalence of infectious diseases in modern society makes the living environment of human beings more severe. It is more likely that we will see more emerging infectious diseases like COVID-19 in the future; these will be widespread, highly prevalent, and harmful [100]. In the face of such a severe test, CRISPR-based detection technology will be a powerful weapon. In the natural environment of population susceptibility screening, more and more infectious diseases are developing in the direction of pathogenic higher evolution, resulting in rapid changes in the nucleic acid sequences of viruses and bacteria. In the face of the evolution of infectious diseases, nucleic-acid-sequence-detection technology based on the CRISPR/Cas system will become the key to preventing pandemics in the future.

## Figures and Tables

**Figure 1 diagnostics-12-02455-f001:**
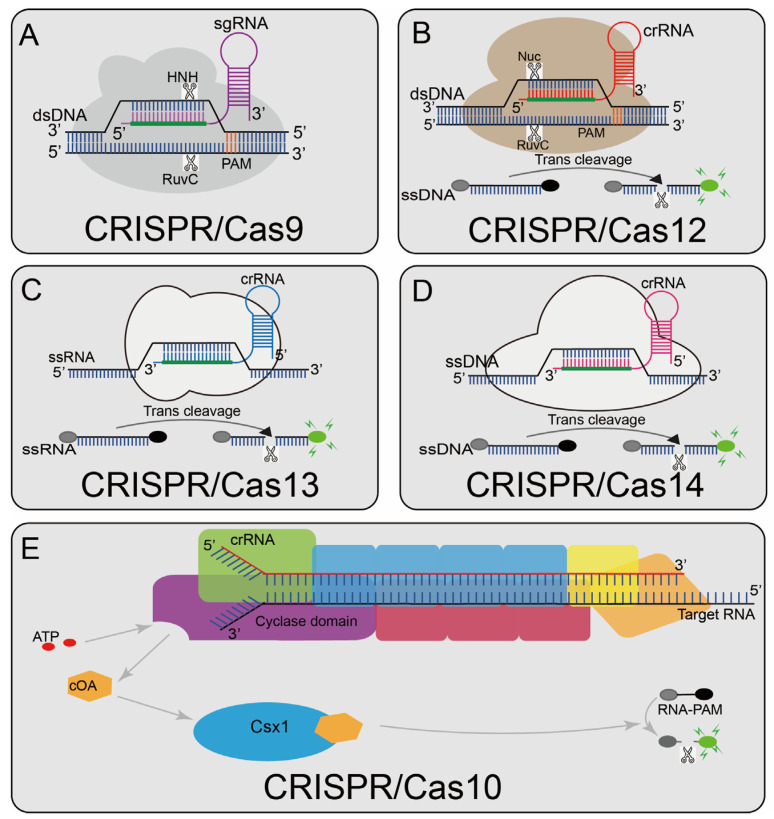
Overview of CRISPR-Cas enzyme activities and their catalytic mechanisms (**A**) Cas9 can cleave the target and non-target strands of DNA; a short trinucleotide PAM is also essential for the initial DNA binding; (**B**) Cas12a can cleave dsDNA under the guidance of gRNA. The Cas12a enzyme recognizes the PAM of the original T-rich spacer and then recognizes the target sequence to generate PAM distal dsDNA breaks with staggered 5′ and 3′ ends, and Cas12 has the side chains trans-cleavage activity. At the time that the sgRNA-guided DNA is combined in Cas12, Cas12 will release a powerful, indiscriminate single-stranded DNA (ssDNA) cleavage activity; (**C**): Cas13 can activate its single-stranded RNA (ssRNA) cleavage activity by binding to crRNA, and it has a additional cleavage activity triggered by the target RNA; (**D**) Cas14 protein is a RNA-guided nuclease and can recognize the target ssDNA without restriction sequences and cleave it, and also can non-specifically cleave the surrounding ssDNA nucleases molecule; (**E**) Cas10 is a multi-component and multi-pronged immune effector that can be activated by viral RNA, and the viral RNA will activate Cas10 polymerization enzymes that produce about 1000 cyclic nucleotides (cOA). cOA activates Csx1, cutting off the fluorophore connected to the quencher.

**Figure 2 diagnostics-12-02455-f002:**
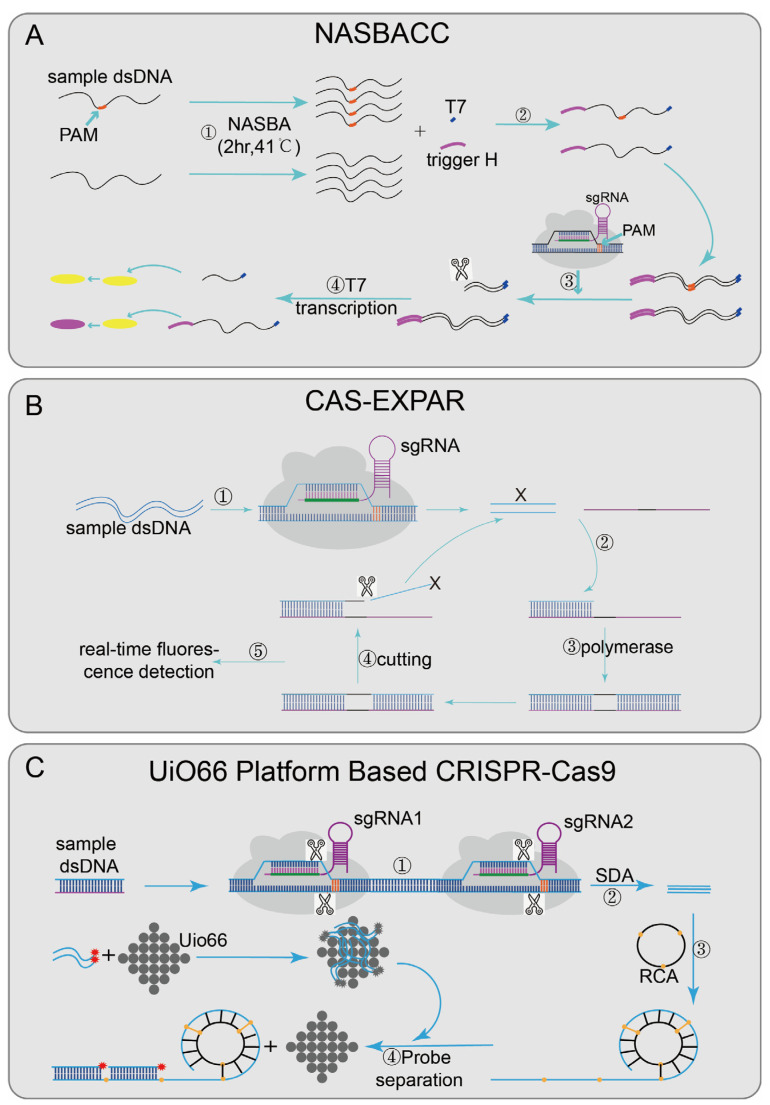
Applications of CRISPR/Cas9 technology: (**A**) NASBACC designed a sgRNA based on the NGG PAM, a specific sensor trigger sequence was added on the amplified product after NASBA amplified, if there is a target sequence, the added sensor sequence will be cut off, and subsequent responses cannot be triggered; otherwise, the added sensor sequence will trigger the sensor response; (**B**) Cas-EXPAR can specifically cleave the target sequence under the guide of the designed sgRNA, produce cleaved fragments (X), X hybridizes to the EXPAR template and is extended along the template by DNA polymerase from its 3′ end. The subsequently formed duplex is cleaved by Nease; a copy of X is released from the template, finally combined with fluorescence intensity analysis to achieve detection; (**C**): UiO66-platform-based CRISPR/Cas9 designed two Cas9/sgRNA complexes that recognize and cleave the target DNA to produce short ssDNA and perform rolling circle amplification; when long ssDNA is present, the fluorescent probe will dissociate from UiO66 and combine it with long ssDNA to regenerate the fluorescent signal; finally, the changed fluorescence intensity is detected to detect the presence of the target nucleic acid.

**Figure 3 diagnostics-12-02455-f003:**
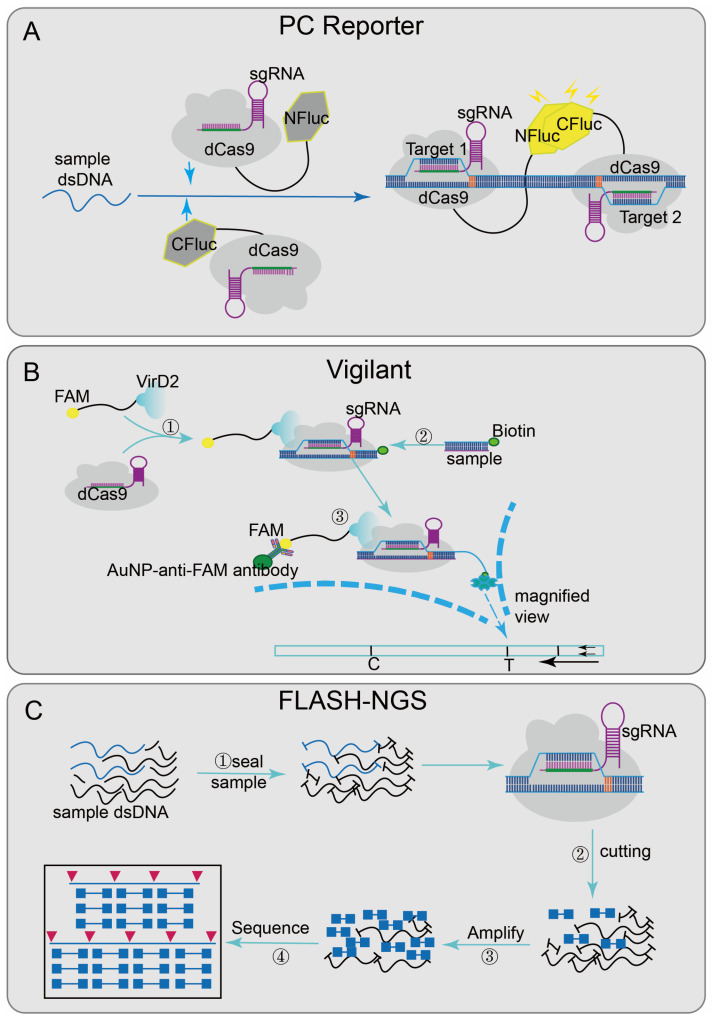
Applications of CRISPR/Cas9 technology: (**A**) PC Reporter has two paired dCas9/sgRNA complexes that are respectively connected to the N-terminal and C-terminal of the firefly luciferase (NFluc and CFluc); when the adjacent sequences are detected in sample, the complementary upstream and downstream segments combined and generated luminescence. (**B**) Vigilant designed a reporting ssDNA with the 5′-end of the 25-BP VirD2 recognition sequence and the 3′-end of the biotinylated sequence; when the target nucleic acid sequence is present, biotin-reporting ssDNA-VirD2-Cas9-sgRNA-targeting ssDNA complexes are formed, and finally the results will be reported by IFA; (**C**): FLASH-NGS, block the sample genomic DNA or cDNA with phosphatase at first and then combined the recombinant Cas9 and multiple-guide RNAs to cleave the sequence of interest into Illumina sequencing-size fragments; through subsequent amplification, the target sequence is enriched in the background and combined with the sequencing flow cell to achieve multiple detection.

**Figure 4 diagnostics-12-02455-f004:**
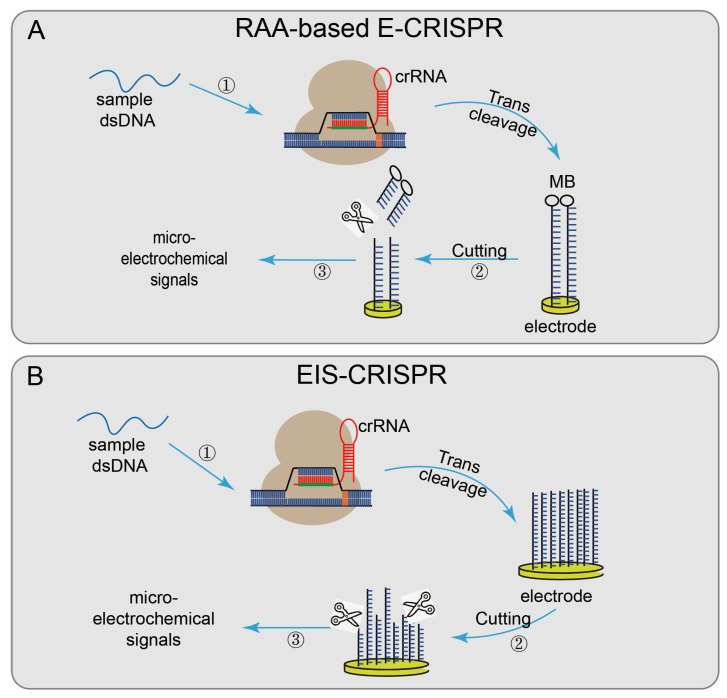
Applications of CRISPR/Cas12technology: (**A**) RAA-based E-CRISPR, uses MB to modify the ssDNA reporter gene and assemble it on the working electrode, the sample is first amplified by RAA, when the target sequence exists, non-specifically cleaves the MB-modified reporter gene on the electrode surface, finally analyzed by SWV to measure the microelectrochemical signal before and after the introduction of the target nucleic acid sequence; (**B**) EIS-CRISPR, fixes ssDNA on a gold electrode to limit the electronic communication between the electrode and the solution; when the target DNA exists, the Cas12/gRNA system binds to the target DNA and trans-cleaves the ssDNA on the gold electrode and accelerates the electron transfer between the electrode and the solution, detecting subtle changes in the electrode surface current at last.

**Figure 5 diagnostics-12-02455-f005:**
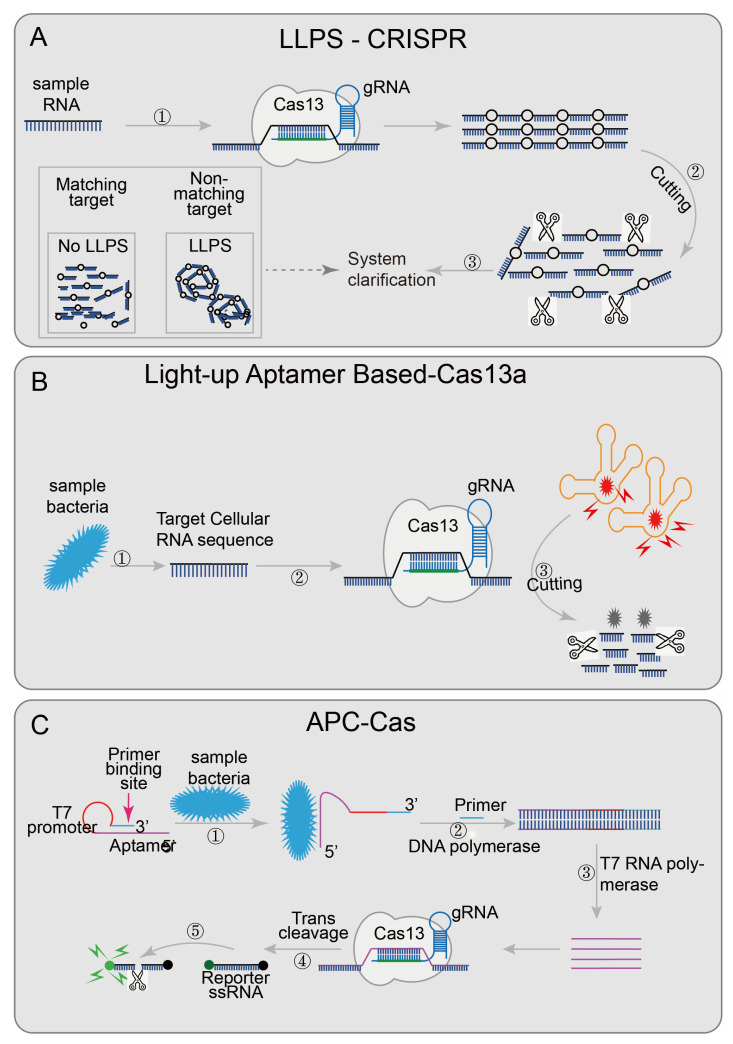
Applications of CRISPR/Cas13 and CRISPR/Cas14 technology: (**A**) LLPS-CRISPR, combined with the collateral cleavage activity of Cas12a/Cas13a, cleaves long-chain into short-chain nucleotides when the target sequence is present; then the solution will become clear afterwards; (**B**) Light-up aptamer-based-Cas13a introduces a new light-up RNA aptamer broccoli/DFHBI-1T complex; when the target sequence is present, Cas13a digests the aptamer broccoli, and the high-fluorescence bound-state DFHBI-1T becomes the low-fluorescence free state; (**C**) APC-Cas’s aptamer domain will specifically recognize and bind to the target pathogen, so that AP expands from a hairpin-like inactive structure and transforms into an active structure; the primer domain can be combined with the primer, and then, with the participation of DNA polymerase, AP is used as the template chain to generate dsDNA, which replaces the target pathogen and realizes the first amplification; then the T7 promoter domain is amplified by T7 RNA polymerase to achieve the second step of amplification; subsequently, the Cas13a/crRNA complex recognizes the ssRNA produced by the second step and non-specifically cleaves a large number of surrounding RNA gene reporter probes, achieving the third step of amplification, finally generating a fluorescent signal.

**Figure 6 diagnostics-12-02455-f006:**
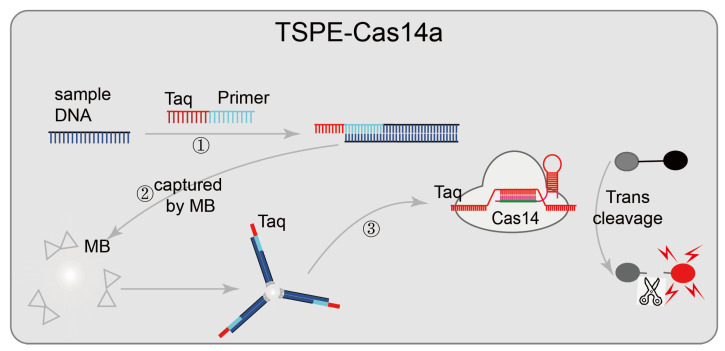
Applications of CRISPR/Cas14 technology: TSPE-Cas14a, designed a tag-specific primer containing two domains (primer sequence domain matching the target, and a tag sequence domain matching sgRNA); after the target sequence is matched, the primer sequence domain begins to be amplified and enriched; then the target nucleic acid is separated from the mixture using streptavidin-coated magnetic beads, and finally, matches sgRNA identifying the tag sequence domain and cleaves the fluorescence quenching reporter gene, resulting in an enhanced fluorescence signal to achieve detection.

**Figure 7 diagnostics-12-02455-f007:**
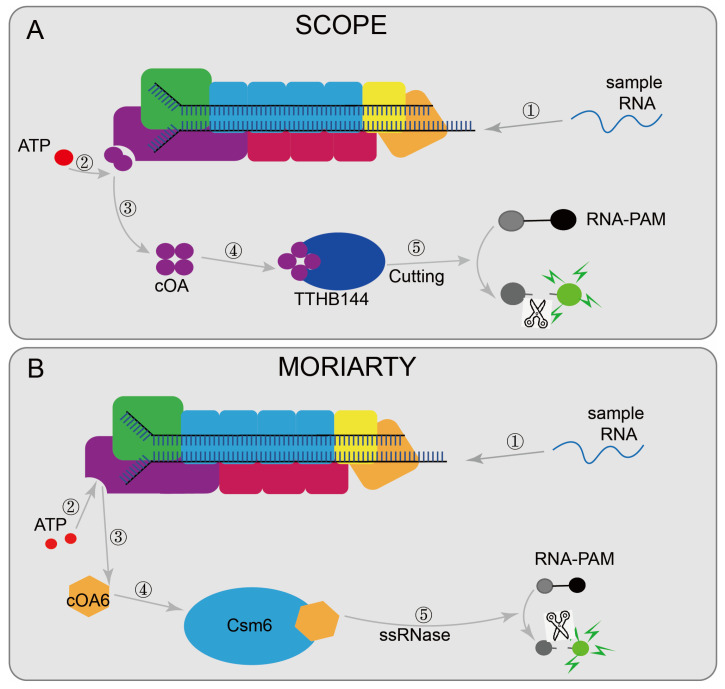
Applications of CRISPR/Cas10 technology: (**A**) MORIARTY, when the target sequence is present, activates Csm1 and synthesizes cOA6 with the participation of divalent ions and ATP, and then cOA6 activates the RNase activity of Csm6, lastly cleaving RNA-FAM; (**B**) SCOPE designed a TtCmr/crRNA complex targeting RNA; after the complex recognizes the target RNA, it can produce cOA molecules, which in turn triggers the cleavage of the reporter RNA by TTHB144, thereby generating a detectable fluorescent signal for result reading.

**Table 1 diagnostics-12-02455-t001:** Major characteristics of various CRISPR/Cas-based biosensing systems ^a^.

Classification	System Name	Effector	Signal Amplification	Sensitivity ^b^	Specificity	Quantitative	Multiplex	Detection Indicator	Detection Time	Target Type	Infectious Disease Target
Cas9-based class	NASBACC [24]	SpCas9	NASBA	fM	1 nt	Y	Y	paper sensors	<4 h	RNA	Zika virus
	CAS-EXPAR [25]	SpCas9	EXPAR	aM	1 nt	N	N	fluorescence	<1 h	DNA	*L. monocytogenes*
	FLASH-NGS [26]	SpCas9	FLASH	NA	NA	N	Y	sequencing	NA	DNA	*S. aureus*
	Vigilant [27]	SpCas9	RT-RPA	2.5 copies	NA	N	N	lateral flow analysis	35 min	DNA	SARS-CoV-2
	UiO66-platform-based Cas9 [28]	SpCas9	RCA	fM	NA	N	N	fluorescence	2 h	DNA	*E. coli* O157:H7
dCas9-based class	PC Reporter System [29]	SpdCas9	PCR	One copy	NA	N	N	fluorescence	PCR + 10 min	DNA	*Mycobacterium tuberculosis*
Cas12a-based class	DETECTR [30]	LbCas12a	RPA	aM	6 nt	N	N	fluorescence	2 h	DNA	HPV
	HOLMES [31]	LbCas12a	PCR	aM	1nt	N	N	fluorescence	NA	DNA	pseudorabies virus and Japanese encephalitis virus
	ITP–CRISPR [32]	LbCas12a	RT-LAMP	10 copies	NA	N	N	fluorescence	35 min	DNA	SARS-CoV-2
	Cas12aFDet [33]	LbCas12a	PCR/RAA	aM	NA	N	N	fluorescence	1 h	DNA	*L*. *monocytogenes*
	OCTOPUS [34]	LbCas12a	RPA	aM	NA	N	N	fluorescence	50 min	DNA	*E. coli* O157:H7 and *Streptococcus aureus*
	Cas12a-UPTLFA [35]	LbCas12a	RPA	pM	1 nt	N	N	fluorescence	80 min	DNA	*Yersinia pestis*
	RAA-based E-CRISPR [36]	AsCas12a	NA	aM	NA	N	N	Current signal	NA	DNA	*L. monocytogenes*
	EIS- CRISPR [37]	LbCas12a	PCR	nM	NA	N	N	Current signal	1.5 h	DNA	*E. coli* and *Staphylococcus aureus*
Cas12b-based class	HOLMESv2 [38]	AacCas12b	LAMP/PCR/RT-LAMP	aM	1 nt	Y	N	fluorescence	1 h	DNA	NA
	CDetection [39]	AaCas12b	RPA	aM	1 nt	N	N	fluorescence	NA	DNA	HPV
Cas13-based class	SHERLOCK [40]	LwCas13a	RPA	aM	1 nt	N	N	fluorescence	2 h	DNA/ RNA	Zika and dengue virus
	APC-Cas [41]	LwCas13a	PCR	1CFU	NA	Y	N	fluorescence	140 min	bacterial pathogen	*Salmonella Enteritidis*
	CARMEN [42]	LwCas13a	PCR/RPA	NA	NA	N	Y	fluorescence	NA	DNA/ RNA	SARS-CoV-2
	ERASE [43]	LwCas13a	RT-RAA	1copy	NA	N	N	Lateral flow strip	60 min	RNA	SARS-CoV-2
	LLPS-CRISPR [44]	LwCas13a	PCR/RPA	aM	NA	N	N	System turbidity	60 min	DNA/ RNA	NA
	Light-up Aptamer based-Cas13a [45]	LwCas13a	NA	10CFU	NA	N	N	fluorescence	NA	bacterial pathogen	*B. cereus*
Cas 14-based class	TSPE-Cas14a [46]	LbCas14a1	PCR	aM	1 nt	N	N	fluorescence	NA	RNA	six pathogenic species
	CMP [47]	LbCas14a1	PCR	aM	NA	N	Y	fluorescence	80 min	RNA	*Streptococcus pyogenes* and *Eberthella typhi*
Cas10 -based class	MORIARTY [48]	LlCsm	RT-RPA	aM	1 nt	N	N	fluorescence	50 min	RNA	SARS-CoV-2
	VmeCmr–NucC coupled assay [49]	VmeCmr	RT-PCR	fM	1 nt	N	N	fluorescence	30 min	RNA	SARS-CoV-2
	SCOPE [50]	TtCmr	RT-LAMP	aM	1 nt	N	N	fluorescence	35 min	RNA	SARS-CoV-2

^a^ All features were claimed by the original publications; CFU, colony-forming units; aM, 10–18 M or attomole/l; fM, 10–15 M or femtomole/l; N, no; NA, not applicable; nt, nucleotide; Y, yes. ^b^ Only the highest sensitivity was shown in the table.

## Data Availability

All data included in this study are identified within the text, tables, and figures.

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
