# Peer review of "Application of CRISPR/Cas Systems in the Nucleic Acid Detection of Infectious Diseases"

_diagnostics, 2022, doi:10.3390/diagnostics12102455_

Round 1
Reviewer 1 Report
The authors summarize recent advances in utilizing CRISPR/Cas9 technology to detect several classes of biomolecules, with an emphasis placed on detecting nucleic acids from pathogens of pandemic concern. This emerging area of research has high therapeutic and diagnostic potential and would be of interest to a broad audience. However, the writing style, limited descriptions and lack of detail provided for each system, and the sheer number of systems described make it very challenging to understand how these systems function and their corresponding utility (over 24 systems are discussed). Both the text and accompanying figure panels lack essential details necessary for understanding how these systems function and how the detection method reports on that functionality. Additionally, many abbreviations used in the text are not defined and the overall writing style/grammar make it very challenging to understand what is being described. Lastly, it would be beneficial to discuss what the strengths/weaknesses or Pros/Cons are for each of these systems to fully appreciate their ability to detect emerging pathogens quickly and accurately.
Major Concerns:
1. Many abbreviations in the text are not defined or are used initially and then defined later in the text. Some examples are included below, but many more are in the text:
- crRNA, used first on line 58, defined on line 63
- dsDNA, used first on line 70 as an abbreviation, then spelled out on line 124
- NASBA, not defined
- EXPAR, not defined
- VirD2, not defined
- SpdCas9, not defined
- FLASH-NGS, not defined
- RAA, used on line 237, defined on line 240
- EIS, used on line 244, defined on line 249
2. In general, many descriptions for the CRISPR-based detection systems are confusing and lack essential details that enable the reader to understand how the systems actually function. As an example, the Cas-EXPAR description states, “It can quickly detect specific nucleic acid sites and methylated DNA levels with a detection limit as low as 0.82 attomolar. Cas-EXPAR uses the target DNA fragments generated by CRISPR/Cas9 cleavage as primers, and the amplification reaction is performed cyclically to generate DNA replicates, which are detected by the real-time fluorescence monitoring method.” How does this detect methylation sites? What generated the fluorescence signal? Is this a qRT-PCR based reaction? What does "the amplification reaction is performed cyclically" mean?
3. The figures accompanying each detection system lack essential details to make it clear to the reader what is happening. For example, Fig. 1B has 6 arrows that point to several different steps in the detection mechanism, yet none are labelled to help the reader understand what is happening between steps. Likewise, the starting materials and products for each step are not labelled…is the figure depicting an RNA molecule, a DNA molecule, an RNA/DNA hybrid? This is a common problem for all figure panels and makes it nearly impossible for the reader to piece together how these systems work.
4. It would be useful to know what the strengths/weaknesses or Pros/Cons are for these systems. Are they all equally reliable? Are they prone to false-positives or false-negatives? Are they cost effective?
Author Response
Point 1: Many abbreviations in the text are not defined or are used initially and then defined later in the text. Some examples are included below, but many more are in the text:
- crRNA, used first on line 58, defined on line 63
- dsDNA, used first on line 70 as an abbreviation, then spelled out on line 124
- NASBA, not defined
- EXPAR, not defined
- VirD2, not defined
- SpdCas9, not defined
- FLASH-NGS, not defined
- RAA, used on line 237, defined on line 240
- EIS, used on line 244, defined on line 249.
Response 1: We are very grateful for your valuable advice. We have re-checked the full text carefully and changed and defined all incorrect or missing abbreviations.
Point 2: In general, many descriptions for the CRISPR-based detection systems are confusing and lack essential details that enable the reader to understand how the systems actually function. As an example, the Cas-EXPAR description states, “It can quickly detect specific nucleic acid sites and methylated DNA levels with a detection limit as low as 0.82 attomolar. Cas-EXPAR uses the target DNA fragments generated by CRISPR/Cas9 cleavage as primers, and the amplification reaction is performed cyclically to generate DNA replicates, which are detected by the real-time fluorescence monitoring method.” How does this detect methylation sites? What generated the fluorescence signal? Is this a qRT-PCR based reaction? What does "the amplification reaction is performed cyclically" mean?
Response 2: Thank you very much for your suggestion. We have revised the unclear points described in the manuscript, and rewritten the confusing points you raised in more detail. The revised manuscript is sent to you as an attachment. The method for real-time fluorescence detection is based on the reaction system of qRT-PCR, and the principle of detecting methylation sites is by combining with bisulfite conversion, that is, after bisulfite treatment, all cytosines on ssDNA are converted to uracil, except for 5-methylcytosine, which has a single base mutation between unmethylated and methylated DNA.U-G mismatches occur during Cas9/sgRNA recognition and DNA polymerase extension when bisulfite-treated unmethylated DNA is recognized in CAS-EXPAR. The 5-methylcytosine was designed at the cleavage site, and the guide sequence of sgRNA and EXPAR template was designed according to bisulfite converted methylated ssDNA sequence, and finally achieved the purpose of detecting methylation. "the amplification reaction is performed cyclically" means to perform the previous cycle again to enrich the template "X", to amplify the fluorescent signal reaction.
Point 3: The figures accompanying each detection system lack essential details to make it clear to the reader what is happening. For example, Fig. 1B has 6 arrows that point to several different steps in the detection mechanism, yet none are labelled to help the reader understand what is happening between steps. Likewise, the starting materials and products for each step are not labelled…is the figure depicting an RNA molecule, a DNA molecule, an RNA/DNA hybrid? This is a common problem for all figure panels and makes it nearly impossible for the reader to piece together how these systems work.
Response 3: We have revised the figure of the full text, and introduced serial numbers such as "â‘ â‘¡â‘¢" to indicate the sequence in which each step occurred, and which change was caused to achieve the detection. This satisfies that readers can directly understand how these systems work through the diagrams we have drawn. The revised diagrams are also attached to the manuscript and look forward to your re-review.
Point 4: It would be useful to know what the strengths/weaknesses or Pros/Cons are for these systems. Are they all equally reliable? Are they prone to false-positives or false-negatives? Are they cost effective?
Response 4: Thank you very much for your suggestions, your questions are very interesting, and knowing the cost-effectiveness of each study will be more helpful for the detection method to move from the research status to the practical application status. Unfortunately, when we reviewed the included studies, we found that most studies were in the preliminary validation stage of the theory and did not give enough data to draw cost-effectiveness, so this evaluation could not be performed for the time being. At the same time, each of the studies we included can achieve a sensitivity of more than 90% at its corresponding detection limit, so its sensitivity is mainly limited by its corresponding detection limit. Finally, we summarize the detection limit of each study in Table 1 and Detection specificity. Thank you again for your valuable advice and look forward to your review.

Reviewer 2 Report
The manuscript “Application of CRISPR/Cas Systems in Nucleic Acid Detection of Infectious Diseases” provides a good overview of CRISPR/Cas-Based Biosensing Systems in infectious disease detection included SARS-CoV-2 detection.
The work is well-structured, well-written and easy to understand. It also addresses a subject that is of great interest in the scientific community. I would suggest the publication.
Author Response
Point 1: The manuscript “Application of CRISPR/Cas Systems in Nucleic Acid Detection of Infectious Diseases” provides a good overview of CRISPR/Cas-Based Biosensing Systems in infectious disease detection included SARS-CoV-2 detection.
The work is well-structured, well-written and easy to understand. It also addresses a subject that is of great interest in the scientific community. I would suggest the publication.
Response 1: Thank you for your favorable comments on our manuscript, thank you very much for your careful review of our manuscript, and thank you sincerely for your enthusiastic work.
Reviewer 3 Report
The review “Application of CRISPR/Cas Systems in Nucleic Acid Detection of Infectious Diseases” by Li et al. gives an overview of various methods based on CRISPR/Cas systems applicable for pathogen detection. It emphasizes the advantages of CRISPR-based biosensing technology as a diagnostic tool and its potential in the fight against infectious diseases.
New diagnostic tools that are fast, sensitive, specific, cost-efficient and easy to use, ideally at the point of need, are of great importance. Although, only few methods are transferred from research state to real-world applications. The review presents various CRISPR/Cas detection systems, but needs a more structured and comprehensive way of description.
A short summary of the CRISPR/Cas mechanism with naming essential components should be added to the introduction. Throughout the manuscript, some abbreviations are not introduced.
The section “Classification of CRISPR Biosensing Systems” gives not enough structured information regarding different classes and types.
Table 1 should contain all references.
Some presented methods are difficult to understand without reading the primary literature. Not all are visualized in the figures, where details are hard to recognize and not all components are included in the legends. Description of methods would benefit from a systematic evaluation and critical assessment of assay performance including e.g. limit of detection, sensitivity, specificity, analysis time, needed sample preparation or pre-amplification and tested samples. Usability in real applications, missing steps and limitations should be pointed out. Subheading would be beneficial.
The section “Application of CRISPR/Cas system in SARS-CoV-2 nucleic acid detection” picks up already described methods without referring to the respective sections. Critical assessment with naming also shortcomings of the methods is still missing. Limitations of CRISPR/Cas as biosensing technology are stated in the section “Perspective”, but these are general and does not compensate for evaluation of each method.
Author Response
Point 1: New diagnostic tools that are fast, sensitive, specific, cost-efficient and easy to use, ideally at the point of need, are of great importance. Although, only few methods are transferred from research state to real-world applications. The review presents various CRISPR/Cas detection systems, but needs a more structured and comprehensive way of description.
Response 1: Thank you very much for your suggestion. We have adopted your suggestion of adding subheadings, including the evaluation of various types of detection systems as separate subheadings, and made corresponding supplements. At the same time, in the background introduction, CRISPR biosensing systems Sections such as taxonomy and the application of CRISPR biosensing systems for SARS-CoV-2 nucleic acid detection have been supplemented accordingly, making our manuscript more structured and informative.
Point 2: A short summary of the CRISPR/Cas mechanism with naming essential components should be added to the introduction. Throughout the manuscript, some abbreviations are not introduced.
Response 2: Thank you very much for your suggestion. We have added an introduction to the CRISPR/Cas mechanism and classification in the introduction, and reviewed and revised the abbreviations of the full text. The revised full text is attached to the attachment.
Point 3: The section “Classification of CRISPR Biosensing Systems” gives not enough structured information regarding different classes and types.
Response 3: We have supplemented the section "Classification of CRISPR Biosensing Systems" and the revised full text is attached in the attachment, we are looking forward to your re-review.
Point 4: Table 1 should contain all references.
Response 4: Thank you very much for your suggestion, we have added all references in table 1.
Point 5: Some presented methods are difficult to understand without reading the primary literature. Not all are visualized in the figures, where details are hard to recognize and not all components are included in the legends. Description of methods would benefit from a systematic evaluation and critical assessment of assay performance including e.g. limit of detection, sensitivity, specificity, analysis time, needed sample preparation or pre-amplification and tested samples. Usability in real applications, missing steps and limitations should be pointed out. Subheading would be beneficial.
Response 5: We have re-described some unclear methods, revised the diagrams, and introduced serial numbers such as "â‘ â‘¡â‘¢" to indicate the sequence of each step and which change is finally achieved. We also labeled the starting materials and products, so that readers can understand how these systems work through the diagrams we have drawn, and the revised diagrams are also attached to the manuscript. By summarizing the corresponding pathogen detection of each method in Table 1, all of them showed high sensitivity and specificity, indicating that the included studies all have usability in practical applications. We have supplemented the ambiguities or important missing steps described in each study. For the limitations of each study, we evaluated the sensitivity, specificity, multi-channel detection, and quantification of each study, and summarized the results in Table 1. We also further list the evaluation of each part of the detection system as a subsection under the premise of the original title to make the structure more complete. The revised full text is attached in the attachment.
Point 6: The section “Application of CRISPR/Cas system in SARS-CoV-2 nucleic acid detection” picks up already described methods without referring to the respective sections. Critical assessment with naming also shortcomings of the methods is still missing. Limitations of CRISPR/Cas as biosensing technology are stated in the section “Perspective”, but these are general and does not compensate for evaluation of each method.
Response 6: Thank you very much for your suggestion. We have revised the section "The Application of CRISPR/Cas System in SARS-CoV-2 Nucleic Acid Detection" according to the order of the article structure and described each different Application of the system in SARS-CoV-2 nucleic acid detection. We added subheadings of the evaluation of each type of method after each type of method according to your suggestion and evaluated each type of method, and we summarized the detection limit and detection specificity of each study in Table 1. Finally, we made an overall description in the "Prospect" section to make the manuscript structure more complete. Thank you again for your valuable advice and look forward to your review.

Round 2
Reviewer 1 Report
All of my comments/concerns have been addressed
Author Response
Point 1: All of my comments/concerns have been addressed.
Response 1: We are very grateful for your valuable advice. Thank you for your favorable comments on our manuscript, thank you very much for your careful review of our manuscript, and thank you sincerely for your enthusiastic work.

Reviewer 3 Report
The presented manuscript improved regarding overall structure, introduction and abbreviations. Although, there was no major revision of the content of the paper. Some descriptions of CRISPR/Cas-based detections systems are still difficult to understand. A critical assessment of single methods stating advantages compared to the gold standard as well as limitations and shortcomings for real applications is mostly missing.
In the figures, visualizing the mechanisms, details were added. This helps to identify the components and reactions steps, but makes the figures crowded and some labels are very small and hard to recognize.
Author Response
Point 1: The presented manuscript improved regarding overall structure, introduction and abbreviations. Although, there was no major revision of the content of the paper. Some descriptions of CRISPR/Cas-based detections systems are still difficult to understand. A critical assessment of single methods stating advantages compared to the gold standard as well as limitations and shortcomings for real applications is mostly missing.
Response 1: Thank you very much for your suggestion. We have modified some methods and legends that are not easy to understand, and marked the processing of each detail more clearly in the figure, to facilitate a better understanding of each detection system; In the evaluation of each subsection, the advantages of each detection method compared with the standard and its shortcomings in practical application are supplemented. The revised full text is attached in the attachment, we are looking forward to your re-review.
Point 2: In the figures, visualizing the mechanisms, details were added. This helps to identify the components and reactions steps, but makes the figures crowded and some labels are very small and hard to recognize.
Response 2: Thank you very much for your enthusiastic work. We have enlarged the scale of each graph to make the graph clearer, and we have also enlarged the small labels to further improve our visualization mechanism. The revised figures have been added to the manuscript, and we are looking forward to your review.
